# FerriTag is a new genetically-encoded inducible tag for correlative light-electron microscopy

Nicholas I. Clarke [iD] [1] & Stephen J. Royle [iD] [1]

A current challenge is to develop tags to precisely visualize proteins in cells by light and electron microscopy. Here, we introduce FerriTag, a genetically-encoded chemically-inducible tag for correlative light-electron microscopy. FerriTag is a fluorescent recombinant electron-dense ferritin particle that can be attached to a protein-of-interest using rapamycin-induced heterodimerization. We demonstrate the utility of FerriTag for correlative light-electron microscopy by labeling proteins associated with various intracellular structures including mitochondria, plasma membrane, and clathrin-coated pits and vesicles. FerriTagging has a good signal-to-noise ratio and a labeling resolution of approximately 10 nm. We demonstrate how FerriTagging allows nanoscale mapping of protein location relative to a subcellular structure, and use it to detail the distribution and conformation of huntingtin-interacting protein 1 related (HIP1R) in and around clathrin-coated pits.

---

[1] Centre for Mechanochemical Cell Biology, Warwick Medical School, Gibbet Hill Road, Coventry CV4 7AL, UK. Correspondence and requests for materials should be addressed to S.J.R. (email: s.j.royle@warwick.ac.uk)

To understand cell biology, we must explore subcellular organization in 3D and locate proteins at high resolution. Correlative light-electron microscopy (CLEM) is a powerful technique to do this, since we can combine the specificity and dynamics of fluorescence light microscopy with the high resolution and cellular context of electron microscopy. A current challenge is to develop tools that allow us to track intracellular events using CLEM. Immunogold labeling has long been used for this purpose, however, pre-embedding immogold electron microscopy (EM) is invasive and its applications are limited[1,2]. More recently, attention has turned to genetically-encoded tags for CLEM.

The ideal tag for CLEM should meet the following criteria: (1) fluorescent and electron dense so that it can be visualized by light and electron microscopy, (2) the electron density should be tightly focused and provide good signal-to-noise ratio so the tag is easily distinguishable from background by EM, (3) genetically encoded so that the cell can be processed in its native state without the need for permeabilization, and (4) non-toxic and non-disruptive so as not to interfere with normal cellular function.

Existing tags do not meet all of these criteria. A popular approach has been to use diaminobenzidine (DAB) to form an electron-dense precipitate either by enzymatic-based polymerization using peroxidase[3,4] or singlet oxygen-based polymerization during photo-oxidation[5,6]. Although these tags allow CLEM, they result in low labeling resolution by EM due to the diffuse nature of the precipitate. Therefore, only proteins situated inside organelles, or those discretely localized at high densities can be successfully visualized[7].

The search for an ideal tag for CLEM has continued towards metal ligand-based tags combined with fluorescent proteins. Metal clusters contain elements that are able to scatter electrons and, if tightly focused, should improve resolution and be readily distinguishable from background. Two such tags are concatenated metallothionein[7,8] and bacterioferritin[9], however each have significant drawbacks in their current form. Use of metallothioneins is limited to high abundance proteins and only minimal ultrastructural information is currently possible[7]. Tagging with bacterioferritin is technically demanding, limited to bacteria, and can lead to aggregation and mislocalization of the target[9].

Human ferritin is a complex of 24 polypeptide subunits of light (FTL) and/or heavy (FTH) chains that form a spherical protein shell with internal and external diameters of approximately 7 nm and 12 nm, respectively. Under iron-rich conditions ferritin is able to store iron (<4300 Fe(III) atoms) as a mineral core and is therefore easily visualized by EM[10]. The electron density of ferritin has been exploited for immunoEM for decades[11,12] and we hypothesized that it could be used as an ideal CLEM tag following some modifications.

Here we introduce FerriTag, a new genetically-encoded inducible tag for CLEM. FerriTag is an engineered ferritin particle that can be acutely attached to a protein-of-interest using rapamycin-induced heterodimerization. We demonstrate how FerriTag can be used to tag single proteins at nanometer resolution. As examples we label several proteins including huntingtin-interacting protein 1 related (HIP1R) which links clathrin-coated membranes to the actin cytoskeleton. We find evidence for different conformational states of HIP1R which depend on its localization at clathrin-coated pits or uncoated parts of the plasma membrane.

## Results

**Design and implementation of FerriTagging.** FerriTagging involves the creation of a ferritin particle (FerriTag) which can be inducibly attached to a protein-of-interest using the FKBP-rapamycin-FRB heterodimerization system (Fig. 1a). To do this,

FerriTag is co-expressed with the protein-of-interest which is fused to FKBP-GFP. FerriTag is untagged FTL and FRB-mCherry-FTH1 transfected at a ratio of 4:1. When rapamycin is added, it induces the heterodimerization of FKBP and FRB domains resulting in the target protein becoming FerriTagged (Fig. 1a).

Our initial attempt to use Ferritin as a tag involved the direct fusion of FTH1 to a protein-of-interest, similar to a bacterial system described previously[9]. Mammalian cells expressing mCherry-FTH1 fused to the mitochondrial targeting sequence of Tom70p were cultured under iron-rich conditions and then visualized by fluorescence microscopy (see Supplementary Note 1). It was clear that this direct fusion resulted in aggregation and mislocalization of mitochondria, most likely due to the multivalent nature of the ferritin molecule. This observation drove us to engineer a novel ferritin tag that could form a ferritin particle independently and then be inducibly added to a protein-of-interest to avoid such aggregation issues. The dilution of FRB-mCherry-tagged FTH1 subunits with untagged FTL subunits is also an important step because aggregation was observed when FRB-mCherry-FTH1 was expressed alone. The outcome of this diluted-tagged version of ferritin, was no aggregation nor mislocalization before or after the addition of rapamycin (see Supplementary Fig. 1).

Our first target protein for FerriTagging was clathrin. HeLa cells expressing FerriTag and clathrin light chain a (LCa) tagged with GFP and FKBP were imaged by fluorescence microscopy (Fig. 1b). These experiments show that, after the addition of rapamycin (200 nM), FerriTagging of clathrin-coated structures occurs within seconds (Fig. 1c). The diffuse FerriTag signal can be seen to specifically decorate clathrin-coated structures throughout the entire cell, with full labeling after 30 s (Fig. 1b and Supplementary Movie 1). We also tested if FerriTagging of clathrin inhibited its endocytic function. To do this, we measured transferrin uptake in cells where clathrin had been FerriTagged and compared them to control cells with no rapamycin treatment (Fig. 1d, e). We found no difference in uptake of transferrin, suggesting that FerriTagging on a short (12 min) timescale does not grossly interfere with clathrin function. Inhibition of CME could readily be detected by pre-treatment of the cells with hypertonic sucrose (0.45 M). With this approach working, we next wanted to observe FerriTagging by EM.

**Visualizing FerriTagged proteins by electron microscopy.** In order to assess whether or not FerriTagging works at the EM level, we used correlative light-electron microscopy (CLEM). Our EM protocol was optimized so that we could easily distinguish FerriTag from background and still be able to see cellular ultrastructure (Fig. 2a). We initially FerriTagged two different proteins in disparate locations: 1) monoamine oxidase (MAO), an outer mitochondrial membrane protein, and 2) clathrin light chain a (LCa), part of the clathrin triskelion (Fig. 2b). Following the addition of rapamycin, FerriTag was confirmed by fluorescence microscopy to specifically label each protein-of-interest rapidly. After fixation and processing, ultrathin resin sections taken from the same cell were imaged by EM. In each case, electron-dense particles of approximately 7 nm diameter could be easily distinguished from background, and the ultrastructure of the cell was visible. We were able to locate particles specifically in the vicinity of mitochondria (in the case of MAO) or clathrin-coated pits and vesicles (in the case of LCa) where each protein-of-interest is localized (Fig. 2b).

Since the FerriTagging CLEM protocol involves preincubating cells in iron-supplemented media and subsequent treatment with rapamycin, we explored if these steps perturbed normal cell biology (Supplementary Note 2). Briefly, we found conditions for

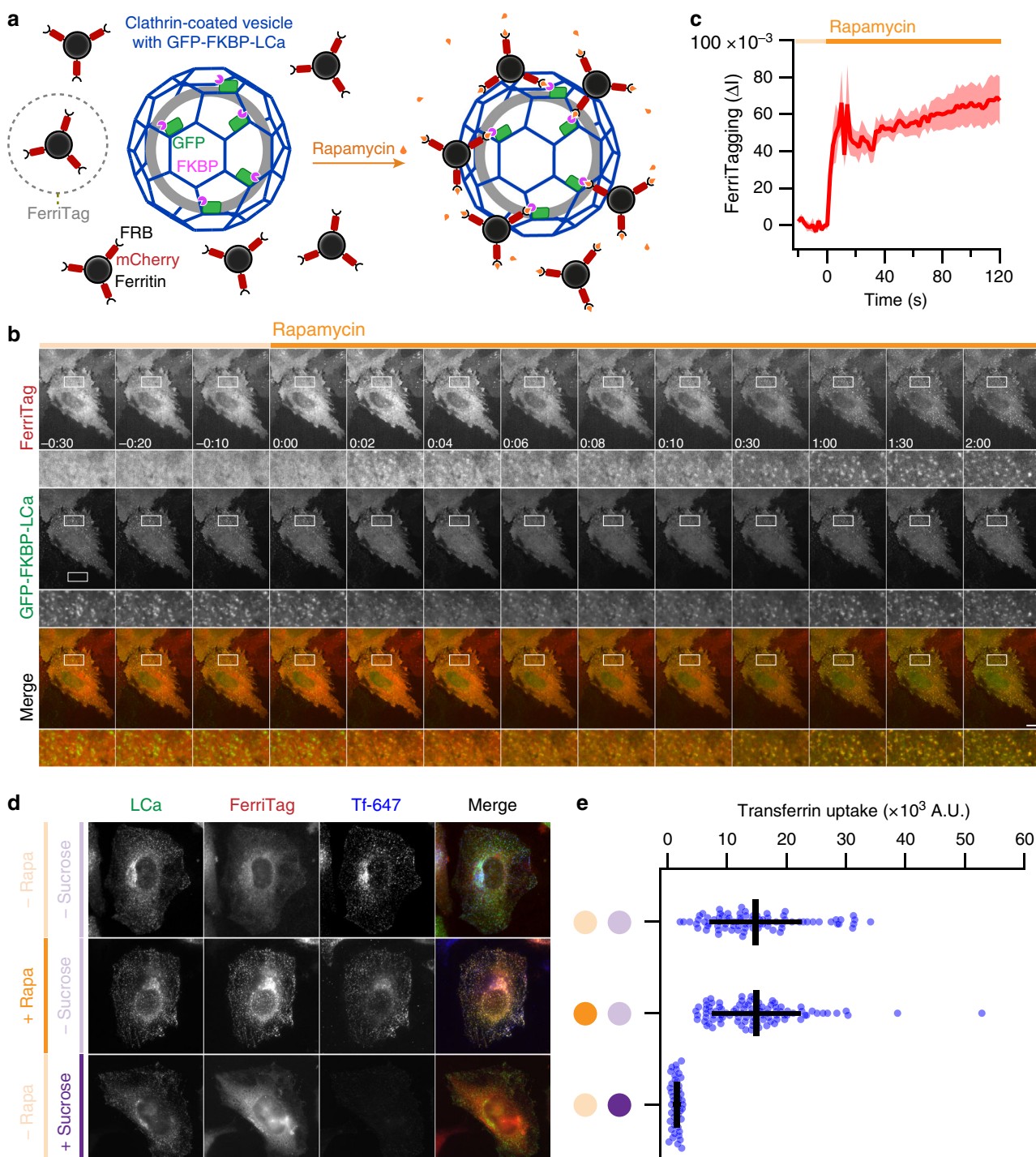

**Fig. 1** Design and implementation of FerriTagging. **a** Schematic diagram of FerriTagging clathrin light chain. Simultaneous expression of FerriTag and GFP-FKBP-tagged clathrin light chain. Addition of rapamycin induces the heterodimerization of FKBP and FRB domains resulting in FerriTagging of clathrin in a clathrin-coated vesicle. **b** Stills from live-cell imaging of FerriTagging clathrin light chain. Rapamycin (200 nM) was added at timepoint zero, as indicated by the orange bar. Specific, rapid labeling of clathrin by FerriTag can be observed immediately. Time, min: sec, scale bar, 10 μm. Zooms show ×4 expansion. See Supplementary Movie 1. **c** FerriTag (mCherry) fluorescence in GFP-FKBP-LCa-positive spots as a function of time. Red trace and shading indicate mean ± s.e.m. for 5 cells. Rapamycin was applied as indicated by the orange bar. **d** Typical images of transferrin (Tf-647, blue) uptake in HeLa cells expressing FerriTag (red) and GFP-FKBP-LCa (green). Rapamycin (200 nM) or control is applied as indicated by dark and light orange bars, respectively. Negative control of hypertonic sucrose to block CME is shown (purple). Scale bar, 10 μm. **e** Scatter dot plot of transferrin uptake. Dots represent measurements from individual cells, from two independent experiments, bars show mean ± SD

iron preincubation in HeLa cells which were non-toxic and did not affect ultrastructure (Supplementary Fig. 2). Iron-loading and rapamycin treatment did not affect the localization of a range of cellular markers representing the actin cytoskeleton, lysosomes, mitochondria, caveolae, endoplasmic reticulum, clathrin coated structures, microtubules, intermediate filaments, nuclei, and adhesion complexes (Supplementary Fig. 3). Incubation of cells with rapamycin for FerriTagging is typically brief, and not

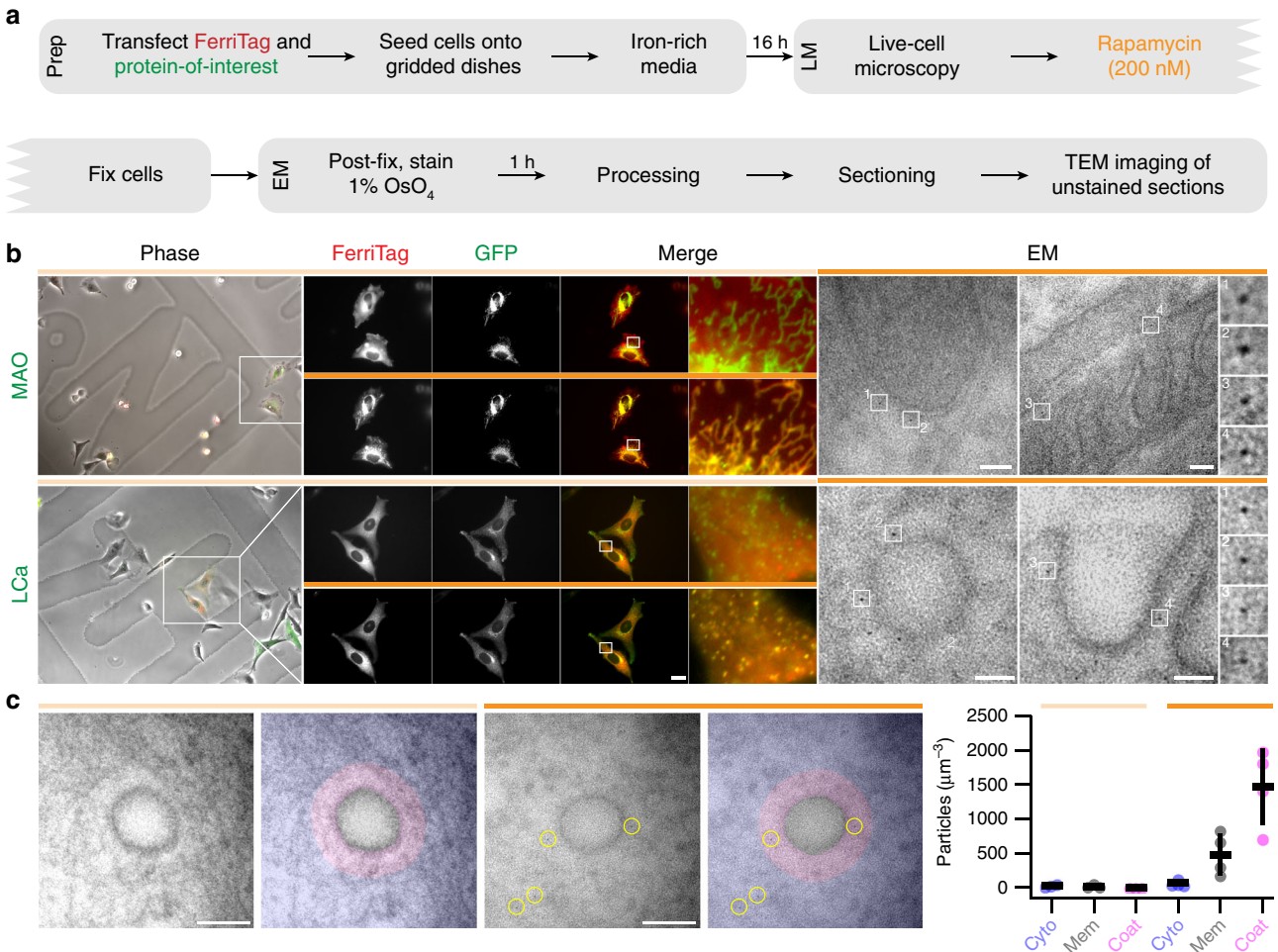

**Fig. 2** Visualizing FerriTagged proteins by light and electron microscopy. **a** Overview of sample preparation steps for correlating light microscopy (LM) with electron microscopy (EM) using FerriTag. **b** Light and electron micrographs of HeLa cells co-expressing FerriTag with either FKBP-GFP-Myc-MAO (MAO) or GFP-FKBP-LCa (LCa). Locator images (left) ensure the same cell can be followed throughout the workflow. Live cell imaging of FerriTagging (middle), cells were treated with rapamycin (200 nM) as indicated by the filled orange bar. The last frame is shown before the cell was fixed and processed for CLEM as described in **a**. TEM images (right) of sections taken from the same cell show FerriTag particles specifically labeling the GFP-FKBP-tagged protein-of-interest in each CLEM experiment. Two particles per micrograph are shown expanded to the right. Light microscopy scale bar 10 μm and zoom ×12. Electron micrograph scale bar 50 nm and zoom ×7.25. **c** Electron micrographs from cells expressing FerriTag and GFP-FKBP-LCa, control (light orange) or rapamycin-treated (dark orange). The location of visible particles, none in control, four in rapamycin-treated are indicated (yellow circles). The incidence of particles in cytoplasmic regions (blue) or coated membrane-proximal regions (pink) was recorded. Membrane-proximal regions are defined as a 50 nm zone on the cytoplasmic side of the membrane. This region is further subdivided into coated areas and areas where no obvious coat is seen. The example micrographs have no uncoated membrane-proximal regions. Scale bar, 100 nm. Right, scatter dot plot to show the density of particles in cytoplasmic (cyto, blue), membrane (mem, gray) or coated (coat, pink) volumes, comparing control ($N_{exp} = 3$) and rapamycin-treated cells ($N_{exp} = 4$). Volumes are calculated by multiplying the area by the section thickness (see Image analysis in Methods)

sufficient to induce autophagy (Supplementary Fig. 4a). A T2098L mutant of FerriTag can be used with the Rapalog AP21967 if prolonged FerriTagging is required and/or autophagy is a concern (Supplementary Fig. 4b).

In Ferritagging experiments, the particles observed by EM most likely correspond to FerriTag for four reasons. First, the size and shape of the particles was consistent with ferritin and labeling was observed in the expected places (Fig. 3a). Second, no labeling was seen at clathrin-coated pits (CCPs) in cells when MAO was FerriTagged, and mitochondria were not tagged in cells when clathrin was FerriTagged (Fig. 3b). Third, no particles were observed next to the appropriate organelle in cells where no rapamycin had been added (Fig. 3c). Fourth, particles were less dense or were absent when FerriTagging was done in cells with no iron loading (Fig. 3d). From these experiments we conclude that

FerriTag works to specifically label proteins-of-interest at the ultrastructural level.

**Efficiency of FerriTagging by electron microscopy.** What are the background concentrations of FerriTag and what densities are seen on the target organelle after FerriTagging? To answer these questions, FerriTag particles in TEM images from cells expressing GFP-FKBP-LCa and FerriTag were counted by an experimenter blind to the conditions (control vs. rapamycin-treated, see Image analysis in Methods). The location of these particles in cytoplasmic or membrane-proximal regions was recorded and the particle densities in each region calculated. Coated areas of the membrane-proximal region could be distinguished and the densities in these areas were also determined.

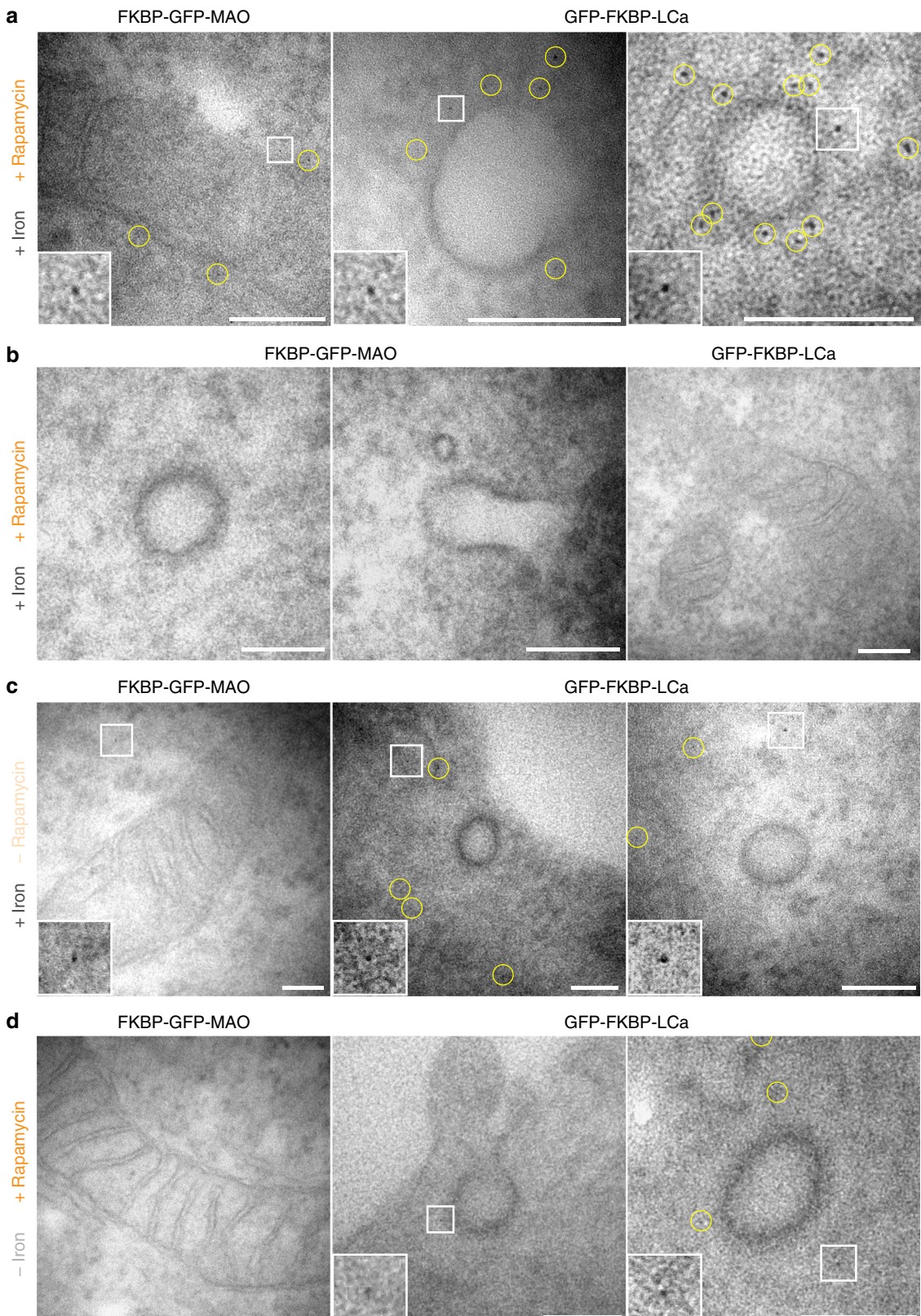

**Fig. 3** Identity of FerriTag particles and specificity of FerriTagging. Electron micrographs from experiments where either FKBP-GFP-MAO or GFP-FKBP-LCa were FerriTagged. Rapamycin addition or iron pre-incubation was done or not as indicated. Electron micrographs show **a** positive controls: labeling of mitochondria in cells where MAO had been FerriTagged (left) and labeling of CCSs in cells where clathrin was FerriTagged (right), **b** internal controls: no labeling of CCSs in cells where MAO had been FerriTagged (left) and no labeling of mitochondria in cells where clathrin was FerriTagged, **c** negative controls: no labeling at the expected locations in the absence of rapamycin, and **d** particles were less dense or appeared to be absent at the expected locations when rapamycin was added but no preloading of iron was carried out. Scale bars, 100 nm

In the control condition, the cytoplasmic vs. coated membrane-proximal density of particles was 34.6 vs. 0 particles per $\mu m^3$ (Fig. 2c). In the FerriTagged samples, the densities were 72.5 vs. 1308.6 particles per $\mu m^3$, a 18-fold enrichment (Fig. 2c). The density of particles in coated membrane-proximal zones in FerriTagged samples was significantly higher than all other densities measured ($p < 0.0022$), all of which were similar to one another ($p > 0.29$, one-way ANOVA with Tukey's post-hoc test).

We used a stereological method to test if the relative labeling intensity (RLI) was different to the expected rate of regular points overlaid onto each image[13]. These points were tested for their coincidence with the cytoplasm or the coated membrane-proximal region. The summary counts and statistics from seven individual CLEM experiments are shown in Table 1. The RLI in the coated membrane-proximal region for rapamycin-treated cells ranged from 2.6 to 4.3 times the expected values, indicating enrichment. The cytoplasmic RLI in rapamycin-treated samples was 0.2 to 0.6 of the expected, suggesting a depletion as particles relocated to membrane regions. For comparison, in controls the cytoplasmic RLI was 1.0 to 1.2 of the expected values. In FerriTagged samples, particles were found at higher than expected rates in membrane-proximal regions with no obvious clathrin coat. Given the specificity of labeling, it is likely that these particles represent FerriTagged clathrin which is in the process of assembly/disassembly. This work indicates that any FerriTag associated with a particular subcellular structure represents real tagging, and that unbound FerriTag does not interfere with detection of genuine FerriTag labeling.

**Determining the labeling resolution of FerriTag.** Labeling resolution refers to the accuracy with which the detected position of the particle corresponds to the location of the labeled protein[2]. The maximum possible distance that the FerriTag particle could be from the tagged protein is theoretically 22 nm (Fig. 4a). However, FerriTag may adopt a more compact conformation or pose resulting in shorter observed lengths. To determine the labeling resolution directly, we FerriTagged the transmembrane protein CD8$\alpha$ and processed samples for EM (Fig. 4b). The perpendicular distance from the center of the FerriTag particle to plasma membrane was measured (median = 9.5 nm, $N = 458$ particles, Fig. 4c). To interpret the shape of this distribution, we carried out computer simulations that modeled the detection of particles in EM sections (see Supplementary Note 3). These simulations indicate that FerriTag can exist in a number of length states from 7 to 18 nm. The distribution has a broad spread (FWHM ≈ 10 nm), which means that on average the particle will be detected $10 \pm 5$ nm away from the target protein. This resolution exceeds that of traditional immunogold labeling which has a labeling resolution of 18 nm in pre-embedding or 21 nm on-section[2]. This dataset also allowed us to directly

determine the signal-to-noise ratio (SNR) for FerriTagging ($8.7 \pm 0.1$, mean ± s.e.m., Fig. 4d, see Supplementary Note 4). The high SNR is encouraging for future computational approaches to automatically pick particles and quantify images from FerriTagged cells.

**Nanoscale mapping of HIP1R using FerriTagging.** Having developed FerriTagging, we next wanted to carry out contextual nanoscale mapping of a protein-of-interest to answer a cell biological question. Huntingtin-interacting protein 1 related (HIP1R), the human homolog of yeast Sla2p, can bind membranes, clathrin light chain, and actin. It exists in two conformations: extended and kinked; and several models have been proposed to explain how HIP1R links the clathrin machinery to the actin cytoskeleton[14–19]. To test these models, we determined the nanoscale distribution of HIP1R at CCPs using FerriTagging. HeLa cells expressing HIP1R-GFP-FKBP and FerriTag were processed through our CLEM workflow and TEM images of FerriTagged HIP1R on CCPs were acquired (Fig. 5a). We collected images of CCPs and segmented the plasma membrane profile and position of FerriTag particles in each (Fig. 5b, see Image analysis in Methods). Using spatial averaging, we plotted the distribution of all particles symmetrically about an idealized pit profile for visualization (Fig. 5c). These data revealed that the distribution of HIP1R is homogenous over the entire crown of the CCP. Moreover, by defining the edges of the CCP we could map FerriTagged HIP1R distal to the CCP, i.e., in adjacent areas of uncoated plasma membrane. Here, HIP1R was labeled at lower density relative to that at the CCP itself (Fig. 5c, d). Interestingly, the distance from particles to the plasma membrane was greater for FerriTagged HIP1R at the pit vs. distal regions, a difference of 10 nm on average (Fig. 5e). FerriTagging is at the C-terminus of HIP1R and any differences in distance to the membrane likely translate into changes in conformation of the molecule (Fig. 5f). These data suggest that HIP1R is in an extended form at the CCP; yet when distal to the pit, HIP1R is in a kinked conformation (Fig. 5f).

## Discussion

In this paper we described FerriTag, a genetically-encoded chemically-inducible tag for CLEM that can be used to acutely label proteins in mammalian cells. The fluorescence and electron density of FerriTag allows proteins to be tracked by fluorescence microscopy in live cells and then visualized at the nanoscale by EM.

FerriTag meets all four criteria for an ideal CLEM tag: (1) fluorescent and electron dense, (2) tightly focused electron density, (3) genetically encoded, and (4) non-disruptive. Currently, the most widely used CLEM tags rely on the production of an electron dense cloud of precipitate that precludes precise

**Table 1 Observed distribution of particles in FerriTag experiments**

| Condition | Expt | $N_{image}$ | $N_{FTo}$ Cyto | $N_{FTo}$ Mem | $N_{FTo}$ Coat | $N_{FTo}$ Total | $N_{FTe}$ Cyto | $N_{FTe}$ Mem | $N_{FTe}$ Coat | $N_{FTe}$ Total | $\chi^2$ | *p-value* |
|---|---|---|---|---|---|---|---|---|---|---|---|---|
| Control | 1 | 26 | 26 | 1 | 0 | 27 | 23.7 | 1.1 | 2.2 | 27 | 0.716 | 0.398 |
| Control | 2 | 29 | 44 | 0 | 0 | 44 | 37.9 | 1.6 | 4.5 | 44 | 2.971 | 0.085 |
| Control | 3 | 6 | 18 | 0 | 0 | 18 | 17.2 | 0.3 | 0.5 | 18 | 0.438 | 0.508 |
| Rapamycin | 1 | 26 | 69 | 7 | 89 | 165 | 123.3 | 7.7 | 34.0 | 165 | 38.531 | $5.39 \times 10^{-10}$ |
| Rapamycin | 2 | 16 | 15 | 15 | 36 | 66 | 50.9 | 3.7 | 11.5 | 66 | 29.301 | $6.20 \times 10^{-8}$ |
| Rapamycin | 3 | 5 | 4 | 1 | 5 | 10 | 7.1 | 1.1 | 1.8 | 10 | 1.092 | 0.30 |
| Rapamycin | 4 | 8 | 7 | 8 | 31 | 46 | 38.1 | 0.6 | 7.3 | 46 | 33.196 | $8.33 \times 10^{-9}$ |

The total number of FerriTag particles observed ($N_{FTo}$) in either the cytoplasm (Cyto), membrane (Mem) or coated membrane (Coat) regions of the indicated number of micrographs ($N_{image}$). The membrane-proximal region comprises Mem + Coat. The expected number of FerriTag particles ($N_{FTe}$) in each region is derived from 24 regular points per image, scaled to the total number of particles observed. The Yates-corrected $\chi^2$ statistic is shown for df = 1 (2 × 2 contingency table—cyto vs. coat, ctrl vs. rapa), the calculated *p-value*

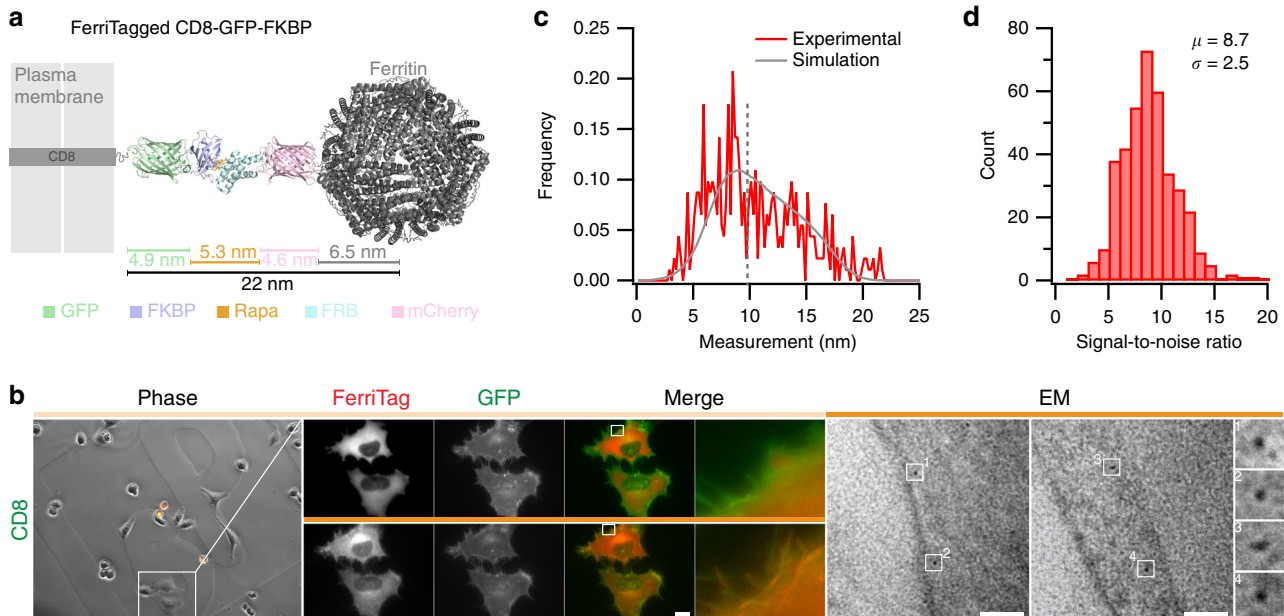

**Fig. 4** Labeling resolution of FerriTagging is approximately 10 nm. **a** Schematic diagram to show the experimental setup to measure labeling resolution of FerriTag. FerriTagging of CD8-GFP-FKBP is shown relative to the plasma membrane. Protein domains are organized co-linearly, giving a total maximum length of 22 nm from the edge of the plasma membrane to the center of the ferritin particle. **b** FerriTagging CD8-GFP-FKBP. Locator images (left), live cell imaging of FerriTagging (middle), and TEM images (right). Cells were treated with rapamycin (200 nM) as indicated by the filled orange bar. Light microscopy scale bar 10 μm and zoom ×12. Electron micrograph scale bar 50 nm and zoom ×7.25. **c** Histogram of experimental observations (red) with a density function of simulated values overlaid (gray). Dotted line indicates the median of the experimental dataset, 9.8 nm ($N_{particle}$ = 458). **d** Histogram of signal-to-noise ratio measured from this dataset ($N_{particle}$ = 389)

localization of target proteins. Due to the tightly focused electron density and good signal-to-noise ratio of ferritin, FerriTag can be used for nanoscale mapping of protein location. There have been previous attempts to use metal ligand-based tags to achieve this[7–9], however there are significant limitations to these methods which have prevented their wide application.

The FerriTag protocol outlined here is simple and robust, with the potential to be used for any protein-of-interest which can be fused to FKBP. Ultrastructure is well preserved due to absence of detergent and it will also possible to incorporate high-pressure freezing into the protocol to better preserve ultrastructure[20], something that is not currently possible with DAB-based genetically-encoded tags. Potential future innovations of Ferri-Tagging include: (1) alternative CLEM protocols that allow for ultra-precise correlated single spot fluorescence localization with FerriTag[21–23], (2) improving staining to enhance the visualization of ultrastructure, (3) combining FerriTag with other tagging methods, perhaps with two-color EM[24], allowing for multicolor EM, and (4) harnessing the magnetic properties of FerriTag for use as a purification tag, or perhaps for direct magnetic manipulation of proteins in living cells. We think FerriTag may also be useful for cryoEM and cryoCLEM applications as the FerriTag particles are likely to provide significant contrast in samples prepared in this way.

FerriTagging has limitations. It is not possible to tag proteins which are located inside organelles, since FerriTag must be able to access the FKBP for FerriTagging to occur. However, successful tagging can be easily assessed by light microscopy, before any samples are processed for EM. Secondly, the iron-loading required for visualizing FerriTag may not be possible in iron-sensitive systems. In these cases it might be possible to enhance the basal iron bound by FerriTag, after fixation during processing.

FerriTagging can be used to track events by light microscopy prior to visualization by EM, but there are two concerns here.

First, does FerriTagging interfere with protein function? We did not observe any adverse effects of FerriTagging on clathrin's endocytic function, suggesting that CCSs could be tracked for long periods. Although any functional impact of FerriTagging of other target proteins would need to be assessed. Second, rapamycin has other effects which may affect live cell imaging. These effects occur on a much longer timescale (tens-of-minutes to hours)[25]. If imaging on this timescale is required, rapalogs may be used together with a mutated form of FerriTag (see Supplementary Note 2). Our results showed that FerriTagging occurs within seconds. For most applications this timescale is sufficient to track events and proceed to EM.

The labeling resolution of FerriTag is 10 ± 5 nm, which exceeds the predicted resolution provided by standard immunogold labeling by either pre-embedding or on-section[2]. Furthermore, FerriTag exceeds the resolution of currently available super-resolution light microscopy methods, with the added benefit that cellular context can be observed in relation to nanoscale localization of the protein-of-interest[23,26]. Note that this comparison is not straightforward because super-resolution imaging of antibody labeling has a spatial resolution of >15 nm, which is then slightly degraded because of what we term here as the labeling resolution[27]; EM has sub-nanometer spatial resolution. These properties make FerriTag ideal for mapping protein distribution at the nanoscale.

Contextual nanoscale mapping of proteins allows investigators to detail the fine distribution of a protein-of-interest in the context of subcellular ultrastructure. In this paper, we used FerriTagging to do contextual nanoscale mapping of HIP1R in HeLa cells. At least three models have been proposed previously to explain how HIP1R distribution allows coupling of CCPs to the actin cytoskeleton[14–16]. First, immunogold labeling in unroofed cells suggested that HIP1R is restricted to the rim of CCPs[14]. Second, HIP1R was proposed to decorate deeply invaginated

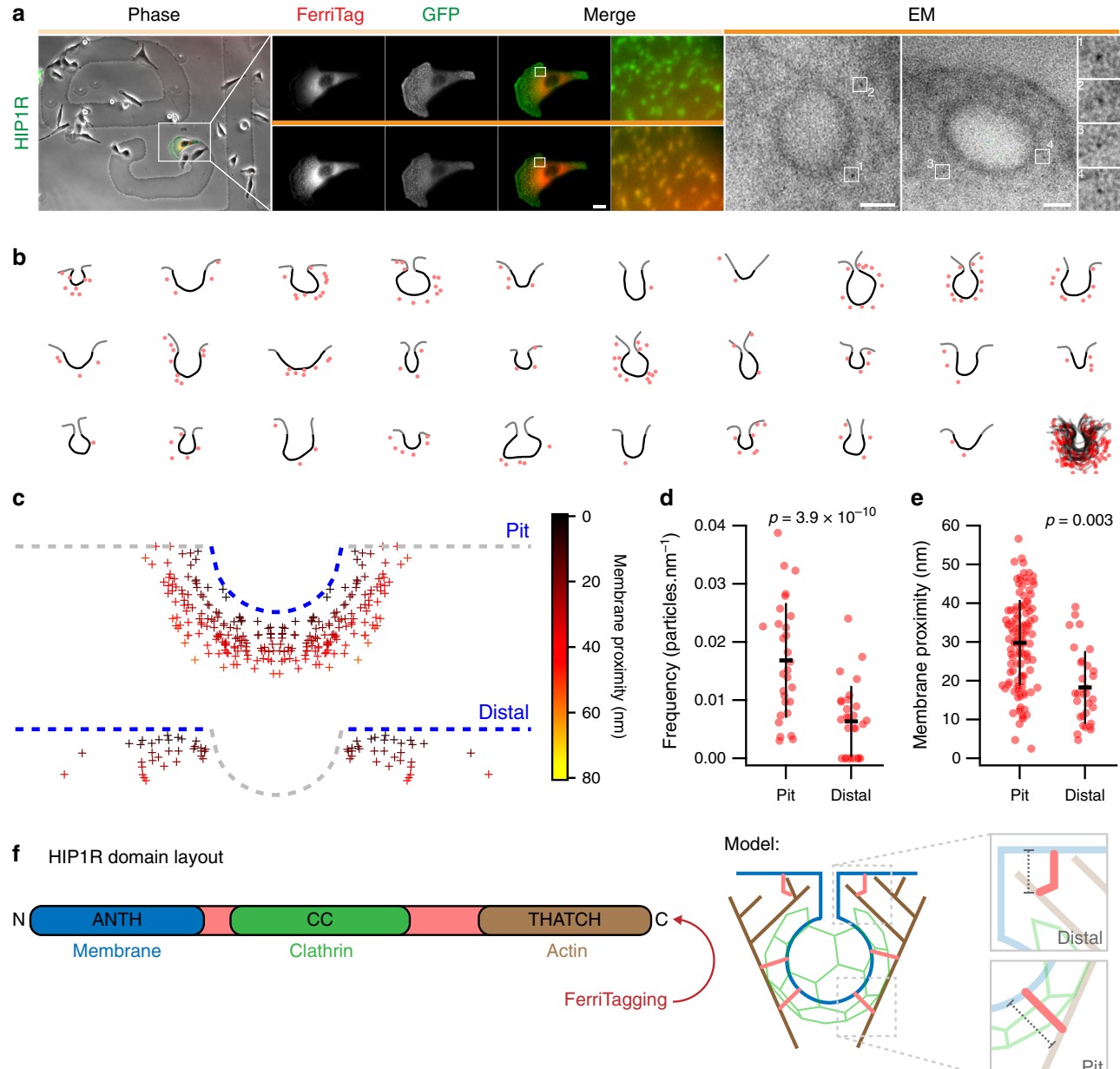

**Fig. 5** Nanoscale mapping of HIP1R in the vicinity of clathrin-coated pits. **a** FerriTagging HIP1R-GFP-FKBP. Locator images (left), live cell imaging of FerriTagging (middle), and TEM images (right). Cells were treated with rapamycin (200 nM) as indicated by the filled orange bar. Light microscopy scale bar 10 μm and zoom ×12. Electron micrograph scale bar 50 nm and zoom ×7.25. **b** Manually segmented membrane profiles (Black—pit, Gray—distal) and FerriTag particles (red) from HIP1R FerriTag electron micrographs. An aligned overlay of all profiles is shown in the bottom right corner. **c** Spatially averaged representation of HIP1R bound to an idealized CCP (100 nm diam.) determined by position of FerriTag particles in TEM micrographs. FerriTag locations are marked by crosses, color-coded by the distance to the plasma membrane. Particles which were closest to the membrane in the pit or in distal regions are presented separately. Note that the distribution is symmetrically presented, i.e., each particle appears twice. **d**, **e** Scatter dot plots of the frequency of FerriTagged-HIP1R (**d**) and the proximity of FerriTagged-HIP1R to the plasma membrane (**e**). Particles in the pit or distal regions are plotted. Bars show mean ± SD, p-values are from Student's t-test with Welch's correction. **f** Schematic diagram of HIP1R, a rod-shaped molecule with interaction domains for membrane, clathrin and actin[14]. Model to explain our data: HIP1R is in an extended conformation when interacting with membrane, clathrin and actin, and is in a shorter/kinked conformation when not bound to clathrin

CCPs towards the neck, where it serves as an anchor against which the polymerizing actin can push to force the vesicle away from the plasma membrane[16]. Note that this model precludes HIP1R in distal regions. Third, HIP1R was suggested to be throughout the pit and in distal regions. Away from the pit, HIP1R was thought to be in an extended strong actin-binding conformation, while in a closed weaker form at the pit itself[15]. In our study, we found that HIP1R is localized throughout the CCP

and in surrounding areas of uncoated membrane, albeit at lower density. HIP1R at the pit appeared to be in an extended conformation whereas in distal regions, the labeling was consistent with a shorter, perhaps kinked, conformation of HIP1R[14]. Our finding of HIP1R throughout the pit is contrary to the first immunogold study, however the unroofing method used there may have removed HIP1R and actin over the crown of the pit, restricting labeling artificially to the rim[14]. In that paper, they also

noted HIP1R immunogold labeling in distal regions, associated with actin filaments[14], which agrees with our findings. The localization we describe is closest to the third model[15], while our conformation data more closely matches the second model[16]. We propose that the N-terminal ANTH domain of HIP1R associates with the plasma membrane and the C-terminal region THATCH domain binds actin[17]. If this is in a distal, uncoated section of plasma membrane the molecule is in a closed conformation. In the clathrin coat, interaction of the middle domain with clathrin encourages the extended conformation of the molecule allowing the coat to link to the actin cytoskeleton[28]. This arrangement allows the coat to be anchored against the cytoskeleton while polymerizing actin serves to push the vesicle away from the plasma membrane. Definitive answers to the conformational state of any protein will require labeling at more than one place in the molecule which should be possible by moving the FKBP domain, provided the protein still localizes and functions normally.

These are exciting times for exploration of the subcellular world and for investigations into protein function in cells at the nanoscale. We hope FerriTag will be widely adopted as a discovery tool on these expeditions.

## Methods

**Molecular biology**. To make FRB-mCherry-FTH1, human ferritin heavy polypeptide 1 (IMAGE clone: 3459353) was amplified by PCR and inserted into pFRB-mCherry-C1 via *XhoI*-*EcoRI*. For expression of FTL only, human ferritin light polypeptide cDNA (IMAGE clone: 2905327) was amplified by PCR and inserted into pEGFP-C1, removing EGFP, via *AgeI*-*XhoI*. Plasmids to express FRB-mCherry-FTH1 and FTL are available from Addgene (100749 and 100750, respectively). Rapalog-compatible FerriTag was made by site-directed mutagenesis to introduce the T2098L mutation into FRB of FRB-mCherry-FTH1. FKBP-GFP-Myc-MAO was a kind gift from Sean Munro (MRC-LMB, Cambridge)[29]. GFP-FKBP-LCa was available from previous work[30]. To make CD8-GFP-FKBP, CD8α was amplified by PCR and inserted into pEGFP-FKBP-N1 via *NheI*-*AgeI*. To make HIP1R-GFP-FKBP, HIP1R (Addgene plasmid 27700) was amplified and inserted into pEGFP-FKBP-N1 via *XhoI*-*AgeI*. Plasmids to express CD8-GFP-FKBP or HIP1R-GFP-FKBP are available from Addgene (100751 and 100752, respectively). To make pMito-mCherry-FTH1, FTH1 was inserted into pMito-mCherry-FRB (from earlier work[30]) via *BsrGI*-*XbaI*. The following plasmids were from Allele Biotech: pmNeonGreen-Actin-C-18, pmNeonGreen-alpha-Actinin-19, pmNeonGreen-Caveolin-C-10, pmNeonGreen-Clathrin-15, pmNeonGreen-H2B-C-10, mNeonGreen-LC3B-7, pmNeonGreen-LAMP1-20, pmNeonGreen-Tubulin-C-35, pmNeonGreen-Vimentin-7, pmNeonGreen-Zyxin-6; whereas GFP-EB1 was available in the lab and pAc-GFP-Sec61beta (Addgene plasmid 15108).

**Cell biology**. HeLa cells were cultured in Dulbecco's Modified Eagle Medium (Invitrogen) supplemented with 10% fetal bovine serum and 100 U/ml penicillin/streptomycin at 37 °C and 5% $CO_2$. Cells were transfected with a total of 1.5 μg DNA (for 3.5 cm dishes) using Genejuice (Novagen) following manufacturer's instructions. The total amount of DNA for each plasmid transfected in FerriTag experiments, unless otherwise specified, was 750 ng for GFP-FKBP tagged protein of interest, 600 ng for FTL only vector and 150 ng for FRB-mCherry-FTH1. Cells were imaged or fixed 2 day after transfection. Cells grown in iron-rich conditions were supplemented with $FeSO_4 \cdot 7H_2O$ to a final concentration of 1 mM in growth media, 16 h prior to imaging. As shown in Supplementary Fig. 2, we determined that this concentration and duration of iron-loading was non-toxic to cells, since 3.3 mM $FeSO_4$ for 72 h was previously shown to alter the ultrastructure of HeLa cells[31]. For transferrin uptake analysis, HeLa cells were serum-starved for 20 min in serum-free DMEM and then exposed to 200 nM rapamycin or ethanol vehicle for 12 min, with 0.05 mg/ml Alexa 647-conjugated transferrin (Invitrogen) added for the final 10 min before fixing. As a negative control, 0.45 M sucrose was applied 15 min at the serum starvation step to inhibit endocytosis. All dilutions were in serum-free media[32].

**Light microscopy**. Live cell imaging of FerriTagging kinetics was performed on a spinning disc confocal microscope (Ultraview Vox, Perkin Elmer) with a ×100 1.4 NA oil-immersion objective at 37 °C. Cells were cultured in glass-bottom fluorodishes (WPI) and kept in Leibovitz L-15 $CO_2$-independent medium supplemented with 10% FBS during imaging. Images were captured using an ORCA-R2 digital CCD camera (Hamamatsu) following excitation with 488 and 561 nm lasers.

Fixed cell experiments were performed in transiently transfected cells attached to cover slips and fixed with 3% paraformaldehyde, 4% sucrose in PBS at 37 °C. Cells were then washed in PBS before being mounted in Mowiol containing DAPI. Imaging was performed on a Nikon Ti epifluorescence microscope with standard filtersets, equipped with a heated environmental chamber (OKOlab) and CoolSnap

MYO camera (Photometrics) using NIS elements AR software. Where applicable, rapamycin (Alfa Aesar) was added by flowing in a concentrated solution in media at 37 °C to a final concentration of 200 nM.

**Correlative light-electron microscopy**. Following transfection, cells were plated onto gridded glass MatTek dishes (P35G-1.5-14-CGRD, MatTek Corporation, Ashland, MA, USA). Light microscopy was performed as described above. Cells were kept at 37 °C in Leibovitz L-15 $CO_2$-independent medium supplemented with 10% FBS during imaging. Transiently expressing cells were located and the photo-etched grid coordinate containing the cell of interest was recorded using brightfield illumination at ×20 for future reference. The same cell was then relocated and fluorescent live cell imaging was acquired at ×100. During imaging, rapamycin was added and once sufficient labeling had been achieved, cells were immediately fixed in 3% glutaraldehyde, 0.5% paraformaldehyde in 0.05 M phosphate buffer pH 7.4 for 1 h. Following fixation, cells were washed several times in 0.05 M phosphate buffer, post-fixed in 1% osmium tetroxide (Agar) for 1 h, washed in distilled water and then dehydrated through an ascending series of ethanol prior to infiltration with epoxy resin (TAAB) and polymerization at 60 °C. This gave sufficient contrast without the need for post-staining. Coverslips attached to the polymerized resin block were removed by briefly plunging into liquid nitrogen. The cell of interest was then located by correlating grid coordinates imprinted on the resin block with previously acquired brightfield images. The resin around the cell of interest was then trimmed away using a glass knife. Serial, ultrathin sections of 70 nm were then taken using a diamond knife on an EM UC7 (Leica Microsystems) and collected on uncoated hexagonal 100 mesh grids (EM resolutions). Electron micrographs were recorded using a JEOL 1400 TEM operating at 100 kV using iTEM software.

**Image analysis**. To measure FerriTagging kinetics in live cell imaging experiments, particle detection was carried out on binarized image stacks. Particles were used as a mask to collect mean pixel densities from both channels. The ratio of background-subtracted intensities was taken for each particle and the median value (set to 0) used for averaging across multiple cells. As a control, the FerriTag channel was randomized and analyzed in the same way, giving no response. Transferrin uptake was analyzed from micrographs by applying a threshold and using Analyze Particles in FIJI[32].

To assess the particle densities, locations of particles in images were manually recorded by an analyst blind to the experimental conditions. In parallel, the images were segmented into extracellular, cytoplasmic, membrane-proximal regions (50 nm zone on the intracellular side of the plasma or vesicular membrane). The membrane-proximal region was further subdivided into areas with a coat and areas where no coat could be detected. A custom-written procedure parsed the list of coordinates of particle locations, classified them according to region and then calculated the volumes of each region in the images using section thickness as a scalar. In addition, according to the methods of Mayhew and Lucocq (summarized in[13]), we tested if the relative labeling intensity (RLI) was different to the expected rate of the intersecting points of an $8 \times 6$ grid overlaid onto the image. The outer lines of the grid coincide with the image perimeter, so $6 \times 4 = 24$ points are tested for their coincidence with the cytoplasmic or the membrane-proximal region. These expected frequencies are scaled to the total number of observed particles for each experiment and tested using Pearson's Chi-squared test with Yates' continuity correction.

Manual detection and measurement of membrane proximity was also used for the CD8-GFP-FKBP dataset. The computer simulations underlying the determination of "labeling resolution" are described in Supplementary Note 3. The signal-to-noise calculations are described in Supplementary Note 4.

For mapping of HIP1R FerriTagging, electron micrographs in TIFF format were imported into IMOD and the plasma membrane and location of electron dense particles was manually segmented. The coordinates corresponding to contours and objects were fed into `IgorPro 7.01` using the output from `model2point`. Custom-written procedures processed these data. First, the coordinates were scaled from pixels to real-world values, and the closest distance (proximity) to the plasma membrane was recorded. Next, the beginning and end of the pit were defined manually in a graphical user interface. The contour length of the pit was determined and the contour length between the start of the pit and the point of closest approach for each FerriTag particle was calculated. The ratio of these two lengths allowed us to plot out a spatially normalized view of the labeling locations. Particles that were closest to the membrane outside of the pit were plotted separately.

**Code availability**. All code used in the manuscript is available (https://github.com/quantixed/FerriTag). Details of the files used and their respective hashes are given in Supplementary Methods.

**Data availability**. All relevant data are available from the authors.

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

## Acknowledgements

We would like to thank Natalie Allcock, Ania Straatman-Iwanowska and Faye Nixon for technical help. We also thank Sean Munro for reagents, and Ian Prior, Anne Straube, Rob Cross and colleagues in CMCB for critical discussion. This work was supported by a Senior Cancer Research UK Fellowship (C25425/A15182) to SJR and a Cancer Research UK Studentship (C25425/A16141).

## Author contributions

N.I.C.: did all experimental work, analyzed data, and wrote the paper, S.J.R.: analyzed data, wrote computer code, and wrote the paper.

## Additional information

**Competing interests:** The authors declare no competing interests.

