## [Peer Review File · Nature Communications]

Editorial Note: This manuscript has been previously reviewed at another journal that is not operating a transparent peer review scheme. This document only contains reviewer comments and rebuttal letters for versions considered at Nature Communications. Mentions of prior referee reports have been redacted.

Reviewers' comments:

Reviewer #2 (Remarks to the Author):

The revised manuscript by Nicholas Clarke and Stephen Royle 'FerriTag: A Genetically-Encoded Inducible Tag for Correlative Light-Electron Microscopy' is improved in several ways. The authors specifically addressed the concern about kinetic of Ferritag labelling to perform timely accurate CLEM. In addition, they proved that Ferritag labelling does not inhibit endocytosis when they labelled clathrin light chain a (LCa). Suggesting that Ferritag labelling does not affect cellular functions, at least in that particular case. The authors also estimated the fraction of protein (clathrin light chain a) labelled upon rapamycin addition.

Nevertheless, I am puzzled by the way the authors quantified the fraction of labelled proteins compared to the control. First it is not clear how many cells were analyzed for the quantification, population of cells or one cell in each case? The authors subtracted the densities of cytosolic and membrane particles quantified in the control from the densities found in the Ferritagged sample. I think that this could not be done, since these densities will vary from cell to cell according to the Ferritag expression level. What could be quantified are changes in densities at the membrane (i.e. the fraction of particles at the membrane). In the control case they found 318 versus 592 in the cytosol versus membrane, thus 65 % (592/910; 1.9 fold) of particles are at the membrane. In the Ferritagged sample they found 348 versus 1493 in the cytosol versus membrane, thus 81 % (1493/1841; 4.3 fold) of particles are at the membrane. So the enrichment increases from 1.9 in the control compare to 4.3 in the Ferritag condition and not 30-fold as said by the authors. According to those results, the real tagging is difficult to estimate. It will represent a little less than 20%. Like suggested in the previous round of revision, it would have been interesting to perform those quantifications after simultaneous permeabilization and fixation, to remove cytosolic FerriTag. Since it is likely that a significant fraction of Ferritag close to the membrane are just cytosolic Ferritag that appear to be at this location during fixation. Could the authors comment on that and change the manuscript accordingly.

I also think that the authors over-interpreted the results obtained in Fig. 4. Their interpretation is that because of HIP1R unfolding the C-terminus of HIP1R is further away from the membrane at the vicinity of the coated pit compared to the base of the invagination (distal regions). To prove this model the authors will need to perform the same experiment with labelled N-terminal ANTH domain, and demonstrate that this domain is close to the membrane at the coated pit. This will be a very convincing demonstration.

At the exception of the points raised above, I am satisfied with answers made by the authors.

Reviewer #3 (Remarks to the Author):

The present paper has undergone a major revision. In the revised version of the manuscript authors have clearly addressed all minor points raised in my review. Overall, the paper is now more clear and well written with improved flow of reading. Moreover, the authors have demonstrated a substantial effort in addressing the comments of others reviewers: additional studies were considered and performed, making the manuscript more experimentally solid. I am convinced that the revised paper fits the scope of Nature Communications journal and I

recommend it for publication.

Reviewer #4 (Remarks to the Author):

This review is based on a previous round, following up on reviewer #1's comments. We assess here if their major issues were addressed.

point 1: the question is whether the strong iron concentrations affect the cell physiology/ultrastructure. The authors only partially address this by showing that the high iron concentrations do not alter cell survival. But no investigation is performed on the ultrastructure. Yet this is critical, especially because the following experiments (labelling density, specificity) rely on the assumption that the ultrastructure is constant.

point 2: the reviewer suggested to use stereological approaches to assess the labelling specificity. This is a very important point that has not been addressed at all. Instead, the authors present an automated image analysis pipeline that does not address the question. A careful consideration of the literature on stereology should be carried on (Gundersen, Mayhew, Lucocq are some important authors in this field) and then applied here.

The quantification showed here does not match the rigorous stereological approaches suggested by the reviewer. cytoplasmic vs membrane labelling has to be evaluated carefully, for example by checking how invariant both compartments are (the notorious reference trap described in many stereological papers). Using the membrane fraction for assessing labelling density is too simplistic. They should measure at endocytic profile to assess the specificity on clathrin. The authors do not describe the sampling strategy for doing the quantification. Unbiased analysis is the key for such quantification. What about the variance?

In a more general consideration, how would one apply this technique when no a priori knowledge of the localization is expected: as the background is significant in the cytoplasm, general application of the technique sounds extremely unlikely or difficult.

Point 3: if rapamycin can induce autophagy, it justifies even more strongly the need for a systematic assessment of the overall sub cellular organisation in control and treated cells. Strengthen the argument of point2. If rapamycin increases the total membranous fraction, then the background labelling on membrane will increase as well.

Point 4: it is very difficult to assess the quality of the EM for technical reasons: absence of contrast is necessary to visualize the small ferritin particles. Moreover, the images show only fractions of the cells. Lower magnifications are expected.

At least the authors should have considered other imaging modalities to fix the detection issue (STEM, HAADF ...).

The answer proposed in this rebuttal therefore sounds too simple and expedite. Moreover, a close look at the image raise again the issue of specificity. Some particles are far away from the membrane profile. at what distance are they considered as cytoplasmic back-ground?

Additional to the previous round:

- As noted by other reviewers, the compromise on the ultrastructure is too high to justify the effort, compared to other CLEM methods or even to immuno-EM

- the throughput of the correlation sounds much less than techniques where FPs fluorescence is preserved in sections (Ref Kukulski papers and follow ups): live cell imaging is necessary, hence only single cells are correlated. Live cell CLEM has an interest to capture transient event at a high temporal resolution but the paradigm presented here is quite tedious and requires long incubations to perform the labelling.

- this method does not fix the major drawbacks of the "Kukulski method", i.e. the need for transfection and over-expression. No localization of endogenous expression is foreseen.

To conclude, Clarke and Royle do not manage to convince of the advantages of this new CLEM approach. The experimental setup seems difficult to adapt to a widespread application, and the claims for a gain in precision and specificity are not supported by the data.

Reviewers' comments:**Reviewer #2 (Remarks to the Author):**

The revised manuscript by Nicholas Clarke and Stephen Royle 'FerriTag: A Genetically-Encoded Inducible Tag for Correlative Light-Electron Microscopy' is improved in several ways. The authors specifically addressed the concern about kinetic of Ferritag labelling to perform timely accurate CLEM. In addition, they proved that Ferritag labelling does not inhibit endocytosis when they labelled clathrin light chain a (LCa). Suggesting that Ferritag labelling does not affect cellular functions, at least in that particular case. The authors also estimated the fraction of protein (clathrin light chain a) labelled upon rapamycin addition.

Nevertheless, I am puzzled by the way the authors quantified the fraction of labelled proteins compared to the control. First it is not clear how many cells were analyzed for the quantification, population of cells or one cell in each case? The authors subtracted the densities of cytosolic and membrane particles quantified in the control from the densities found in the Ferritagged sample. I think that this could not be done, since these densities will vary from cell to cell according to the Ferritag expression level. What could be quantified are changes in densities at the membrane (i.e. the fraction of particles at the membrane). In the control case they found 318 versus 592 in the cytosol versus membrane, thus 65 % (592/910; 1.9 fold) of particles are at the membrane. In the Ferritagged sample they found 348 versus 1493 in the cytosol versus membrane, thus 81 % (1493/1841; 4.3 fold) of particles are at the membrane. So the enrichment increases from 1.9 in the control compare to 4.3 in the Ferritag condition and not 30-fold as said by the authors. According to those results, the real tagging is difficult to estimate. It will represent a little less than 20%. Like suggested in the previous round of revision, it would have been interesting to perform those quantifications after simultaneous permeabilization and fixation, to remove cytosolic FerriTag. Since it is likely that a significant fraction of Ferritag close to the membrane are just cytosolic Ferritag that appear to be at this location during fixation. Could the authors comment on that and change the manuscript accordingly.

The reviewer is absolutely correct. The problem was that our automated particle detection software was not ready for use in this paper. As the reviewer noted, the software was prone to error, especially close to membranes. We were quite wedded to the software having spent a lot of time developing it, and because it is an unbiased way of assessing the data. However, during the revisions we took a step back and came to the conclusion that much more development is needed to get a truly automated analysis pipeline to work on this problem. On top of this, the way we presented the data was quite confusing (also pointed out by the reviewer). So, to remedy this situation and to take into account the comments of the other reviewers, we have undertaken a manual stereological analysis of FerriTag distribution after clathrin has been FerriTagged. Importantly we have also repeated these experiments several times such that we now have a really good dataset to answer the original concern: does background FerriTag interfere with detecting a protein-of-interest.

Briefly, an observer blind to the experimental conditions manually recorded particle locations in all images. A computer program determined whether these particles were in the cytoplasm or a membrane-proximal region. The program also measured the volumes of each region in each image, giving us a density of particles per unit volume for each

region. This analysis is now shown in **Figure 2C**. It shows that there is much more labelling in membrane-proximal regions in FerriTagged samples compared to controls. The density of detected particles in the cytoplasm is very low. This analysis makes the point that “real labeling” is readily distinguishable from background.

We went further and used a stereological method to compare the frequency of particles in each region with that expected by a regular grid of intersecting points. This method was quite sensitive and allowed us to see that the relative labelling density in the cytoplasm is ~ 1 in controls but is reduced in FerriTagged samples to 0.2 – 0.6 (**Table 1**). This is caused by the relocation to the membrane-proximal zone but makes more sense compared to our previous analysis.

I also think that the authors over-interpreted the results obtained in Fig. 4. Their interpretation is that because of HIP1R unfolding the C-terminus of HIP1R is further away from the membrane at the vicinity of the coated pit compared to the base of the invagination (distal regions). To prove this model the authors will need to perform the same experiment with labelled N-terminal ANTH domain, and demonstrate that this domain is close to the membrane at the coated pit. This will be a very convincing demonstration.

It is true that to definitively nail the conformational states of HIP1R we would need to tag it somewhere else in the molecule. Unfortunately, an N-terminal tag, apparently interferes with the HIP1R ANTH domain and therefore membrane-binding. It might be possible to place an FKBP internally in HIP1R, after the ANTH domain and before the coiled-coil to do this experiment, but we focused on the other revision experiments rather than attempting this. We accept that the criticism that these data are stretching the interpretation. We have therefore tempered the way that the results are discussed. A sentence on p. 11 now reads: *Definitive answers to the conformational state of any protein will require labeling at more than one place in the molecule which should be possible by moving the FKBP domain, provided the protein still localizes and functions normally.*

At the exception of the points raised above, I am satisfied with answers made by the authors.

We thank the reviewer for their thoughtful comments and their time reviewing our paper.

Reviewer #3 (Remarks to the Author):

The present paper has undergone a major revision. In the revised version of the manuscript authors have clearly addressed all minor points raised in my review. Overall, the paper is now more clear and well written with improved flow of reading. Moreover, the authors have demonstrated a substantial effort in addressing the comments of others reviewers: additional studies were considered and performed, making the manuscript more experimentally solid. I am convinced that the revised paper fits the scope of Nature Communications journal and I recommend it for publication.

We thank the reviewer for their support for our work and for their comments.

Reviewer #4 (Remarks to the Author):

This review is based on a previous round, following up on reviewer #1's comments. We assess here if their major issues were addressed.

point 1: the question is whether the strong iron concentrations affect the cell physiology/ultrastructure. The authors only partially address this by showing that the high iron concentrations do not alter cell survival. But no investigation is performed on the ultrastructure. Yet this is critical, especially because the following experiments (labelling density, specificity) rely on the assumption that the ultrastructure is constant.

We performed two types of experiment to address the concern that iron-loading may alter ultrastructure. First, we have used SBF-SEM to serially section through cells that were iron loaded (1 mM FeSO₄, 16 h) and compare to those that were not. This allowed us to look broadly at the ultrastructure of whole cells. We could not see any obvious signs of changes in ultrastructure (**Reviewer Figure 1**). Second, we tested whether iron-loading and rapamycin-treatment had any effect on cellular structures by light microscopy. We found no disruption of the actin cytoskeleton, lysosomes, mitochondria, caveolae, endoplasmic reticulum, clathrin coated structures, microtubules, intermediate filaments, nuclei and adhesion complexes. This data is shown in **Supplementary Figure 4**. It is difficult for us to prove a negative, but the fact that there is no change in cell viability and that we see no changes in cellular structures by LM or by EM tells us that our FerriTag protocol is not disruptive, at least for HeLa cells incubated for 16 h in 1 mM FeSO₄.

Reviewer Figure 1: Serial Block Face Scanning Electron Microscopy of HeLa cells cultured overnight with or without iron supplementation.

point 2: the reviewer suggested to use stereological approaches to assess the labelling specificity. This is a very important point that has not been addressed at all. Instead, the authors present an automated image analysis pipeline that does not address the question. A careful consideration of the literature on stereology should be carried on (Gundersen, Mayhew, Lucocq are some important authors in this field) and then applied here.

The quantification showed here does not match the rigorous stereological approaches suggested by the reviewer. cytoplasmic vs membrane labelling has to be evaluated carefully, for example by checking how invariant both compartments are (the notorious reference trap described in many stereological papers). Using the membrane fraction for assessing labelling density is too simplistic. They should measure at endocytic profile to assess the specificity on clathrin. The authors do not describe the sampling strategy for doing the quantification. Unbiased analysis is the key for such quantification. What about the variance?

We thank the reviewer for their input on this point. In the previous revision we wrote software which compared the labelling densities of FerriTag in an unbiased way. We felt that this was a superior approach to classic stereological methods, which predate computational image analysis. We were pretty wedded to this approach having spent several months developing it. The reviewers' critique really made us step back and assess whether the software was fit for purpose. We now realize that it wasn't. We think it can be improved and will be useful eventually, but it is not ready for publication yet.

We have carefully considered the stereological literature and used a method described by Mayhew and Lucocq to assess the relative labelling intensity of FerriTag in cytoplasmic and membrane-proximal compartments. Moreover, we have repeated this analysis in seven separate experiments (three control, four rapamycin-treated) to generate a really good dataset to answer this question. We present a straightforward density analysis in the main paper (**Figure 2C**). The most obvious change is that now we are picking particles manually, we have many fewer particles (note that the quantifier is blind to the experimental conditions). The background frequency of FerriTag is very low and the labelling at the target structures is very high. We used the intersection of internal points in a 8 x 6 grid (24 points) to calculate the expected frequency at each cellular location (cytoplasm vs membrane-proximal). These results clearly show a shift in localization with FerriTagging (**Table 1**). This unbiased analysis using a classical stereological method is a huge improvement to the paper and we thank the reviewer for their persistence on this point.

We have not looked at further restricting the membrane zone into coated regions and non-coated regions. The new analysis is clear enough and is conservative by comparing frequencies in a membrane-proximal region, without potentially adding bias into the analysis.

In a more general consideration, how would one apply this technique when no a priori knowledge of the localization is expected: as the background is significant in the cytoplasm, general application of the technique sounds extremely unlikely or difficult.

It is true that the most straightforward application of FerriTag will be to localize proteins whose approximate location is already known. The reviewer is perhaps overlooking that the fact that this is a CLEM method and so the approximate localization of tagging is seen first by fluorescence microscopy. The scenario where the experimenter is blindly looking for FerriTag is unlikely. The stereological analysis now tells us that the background labelling is very low such that an experiment like the reviewer suggests would be possible and that labelling will be detectable at sites without *a priori* information. We underline that it is unlikely that investigators will do this in the absence of subcellular guidance from light microscopy.

Point 3: if rapamycin can induce autophagy, it justifies even more strongly the need for a systematic assessment of the overall sub cellular organisation in control and treated cells. Strengthen the argument of point2. If rapamycin increases the total membranous fraction, then the background labelling on membrane will increase as well.

Our automated detection approach (now removed from the paper) erroneously detected particles close to the membrane and so this concern has been addressed by the removal of our automated method. As described above and in **Table 1**, the background labelling on membranes is negligible as assessed by stereological analysis.

The deeper question about autophagy induction by rapamycin is whether or not it is a concern and if so, what can we do about it. We thought it was worth pursuing this point as part of the revisions. We show that incubations of up to 10 min with rapamycin (200 nM) do not induce autophagy, as judged by formation of LC3 puncta (**Supplementary Figure 5A**). We think most FerriTag users will fix the cells within this timeframe. However, if longer live cell imaging experiments are required with FerriTagged proteins, we now show that a mutant form of FerriTag (T2098L) can be attached to proteins using the rapalog AP21967 (**Supplementary Figure 5B**). This works similarly to rapamycin but cannot bind mTOR and therefore does not induce autophagy. We think that these new data provide solutions for investigators concerned about autophagy induction.

Point 4: it is very difficult to assess the quality of the EM for technical reasons: absence of contrast is necessary to visualize the small ferritin particles. Moreover, the images show only fractions of the cells. Lower magnifications are expected.

We have now included some lower magnification views in **Supplementary Figure 2**.

At least the authors should have considered other imaging modalities to fix the detection issue (STEM, HAADF ...).

We have considered alternatives to TEM. We have tried to use EELS imaging to locate the FerriTag particles in STEM images. These approaches may ultimately prove successful but they require much more development time and since the particles can be accurately mapped under the conditions in our paper, we have not followed this line further.

The answer proposed in this rebuttal therefore sounds too simple and expedite. Moreover, a close look at the image raise again the issue of specificity. Some particles are far away from the membrane profile. at what distance are they considered as cytoplasmic background?

It is now stated in the paper more clearly that the membrane-proximal zone is a 50 nm region on the cytoplasmic face of the membrane. This accounts for a clathrin density that is ~35 nm from the membrane plus 10 + 5 nm for the labelling resolution of FerriTag.

Additional to the previous round:

- As noted by other reviewers, the compromise on the ultrastructure is too high to justify the effort, compared to other CLEM methods or even to immuno-EM

The disadvantages of other CLEM methods, such as the fact that they are only useful for high density proteins, and that localization is only approximate were set out in the Introduction and the Discussion. Immuno-EM is a wonderful method but, as the reviewer knows it relies on good antibodies to work, so its use is limited.

- the throughput of the correlation sounds much less than techniques where FPs fluorescence is preserved in sections (Ref Kukulski papers and follow ups): live cell imaging is necessary, hence only single cells are correlated. Live cell CLEM has an interest to capture transient event at a high temporal resolution but the paradigm presented here is quite tedious and requires long incubations to perform the labelling.

We do not agree with this comment. The “long incubations” refers to a single overnight step which adds no additional time to an experiment with APEX or the equivalent. By contrast to APEX or other CLEM probes, no further processing to produce the signal after fixation is required, so it is actually faster than those methods. To call our method “tedious” is unfair: any method that allows nanoscale mapping of individual proteins is going to require some effort.

- this method does not fix the major drawbacks of the "Kukulski method", i.e. the need for transfection and over-expression. No localization of endogenous expression is foreseen.

We are not responsible for drawbacks of other peoples’ methods nor do we claim to solve all the weaknesses of alternative methods. The reviewer is incorrect to state that “No localization of endogenous expression is foreseen”. We are currently using FerriTag to label endogenous proteins. We do this by introducing a GFP-FKBP tag at the endogenous locus in cells and using FerriTag as described in our paper (**Reviewer Figure 2**). The tagging of endogenous proteins works similarly to the over-expressed forms in the paper.

Reviewer Figure 2: Micrographs showing labeling of LCa-FKBP-GFP by FerriTag in HeLa cells where FKBP-GFP has been introduced at both alleles of clathrin light chain α . Cells were made and experiment was carried out by Ellis Ryan in our lab. Scale bar, 10 μm .

To conclude, Clarke and Royle do not manage to convince of the advantages of this new CLEM approach. The experimental setup seems difficult to adapt to a widespread application, and the claims for a gain in precision and specificity are not supported by the data.

It's a shame that the reviewer feels this way. We hope that they are not right. At least our group continues to use FerriTagging because we find it useful and because it has benefits over existing techniques. In the end it doesn't really matter. The proof of whether the community finds FerriTagging useful will only be apparent a few years after the publication of this paper. We have distributed the plasmids to several labs already and the preprint has been viewed many times (1,919 abstract views, 902 PDF downloads at the time of writing) so we have good reason to believe that the method will be used.

Reviewers' comments:

Reviewer #2 (Remarks to the Author):

I am completely satisfied with the answers and modifications made by the authors. I think that the development of this new tool for CLEM will be very valuable to the cell biology community.

Reviewer #4 (Remarks to the Author):

In this third round of revision, Clarke and Royle made substantial efforts to address most of the reviewers' comments, especially in reconsidering completely the quantification of Ferritagged particles in order to assess the signal specificity. I appreciate the use of stereology to do this. Nevertheless, when looking at the experiments on tagged clathrin, I do not understand why they measured the tag density at the plasma membrane as a whole and not at the coated structures only. It is obvious that the signal should be restricted to the coated pits or vesicles and not, or significantly less at the plasma membrane. The authors choose the RLI to assess specificity, but they should have better defined their compartment, e.g. compartment 1: coated structures, cpt2: plasma membrane (uncoated), cpt3: cytoplasm. It is even more surprising they did not do this for clathrin, since in the last experiments, they indeed differentiated the coated membranes from the uncoated ones (HIP1R tagging) to characterize (nicely by the way) a differential distribution and potential folding of the protein at the edges of the endocytic pits.

What is not clear, still on the figure 2, is how the authors managed to measure the volume density, i.e. N of particles per cubic micrometers. Have they extrapolated from the section thickness?

These would represent the last major concerns I would like to raise considering the work in its current status.

Moreover, Clarke and Doyle have extended the characterization of the potential adverse effects of the procedure on cellular integrity by checking the distribution of multiple compartment by fluorescence microscopy. Which is convincing.

I can also acknowledge the fact that in the discussion, they address in a transparent manner the drawbacks of the technique, ie lack of contrast, impossibility to label luminal part of organelles, iron sensitivity of some systems etc ... Nevertheless, they could have also highlighted the potential of the in cryoEM and in cryoCLEM in particular as the particles would certainly provide significant contrast.

Other minor points:

- In the introduction, the authors unnecessarily and wrongly criticize the invasiveness of immunolabelling. While it is clearly true for pre-embedding, it is not the case for post-embedding techniques. They should make this clearer in order to avoid defensive reaction of those readers that are still doing iEM.
- As the authors cleverly chose clathrin as a model to prove their labelling technique, they could extend the description in giving an estimate of the labeling efficiency: can they extract this from the predicted number of site at a clathrin pit for example? I believe the stoichiometry is known from previous studies. It would be extremely interesting to estimate how many putative site (LCa) in a section and compare with the number of particles they detect.

Reviewer #4 (Remarks to the Author):

In this third round of revision, Clarke and Royle made substantial efforts to address most of the reviewers' comments, especially in reconsidering completely the quantification of Ferritagged particles in order to assess the signal specificity. I appreciate the use of stereology to do this. Nevertheless, when looking at the experiments on tagged clathrin, I do not understand why they measured the tag density at the plasma membrane as a whole and not at the coated structures only. It is obvious that the signal should be restricted to the coated pits or vesicles and not, or significantly less at the plasma membrane. The authors choose the RLI to assess specificity, but they should have better defined their compartment, e.g. compartment 1: coated structures, cpt2: plasma membrane (uncoated), cpt3: cytoplasm. It is even more surprising they did not do this for clathrin, since in the last experiments, they indeed differentiated the coated membranes from the uncoated ones (HIP1R tagging) to characterize (nicely by the way) a differential distribution and potential folding of the protein at the edges of the endocytic pits.

We didn't make this distinction previously due to being conservative. This is a good suggestion and we have now done this. Most of our images didn't have a lot of uncoated membrane regions so this reclassification hasn't changed the results much. However, it does allow us to now say definitively that the FerriTagging is specific to coated regions. We can also see that the densities in coated regions are higher than in non-coated regions of the membrane proximal zone. We have added the caveat that labelling in the uncoated regions is probably clathrin which is either assembling or disassembling. The specificity of labelling revealed by stereology allows us to make this statement. We have updated:

- *Table 1*
- *Figure 2C scatter dot plot and accompanying legend*
- *The appropriate section in the Results*

We are very grateful to the reviewer for pushing us towards this approach and for their useful suggestions which have improved this part of the paper.

What is not clear, still on the figure 2, is how the authors managed to measure the volume density, i.e. N of particles per cubic micrometers. Have they extrapolated from the section thickness?

This is correct. This was declared in the Methods, although it was not very visible. We have added a note in the figure legend.

Volumes are calculated by taking the area and multiplying by the section thickness (see Methods).

These would represent the last major concerns I would like to raise considering the work in its current status.

Moreover, Clarke and Doyle have extended the characterization of the potential adverse effects of the procedure on cellular integrity by checking the distribution of multiple compartment by fluorescence microscopy. Which is convincing.

I can also acknowledge the fact that in the discussion, they address in a transparent manner the drawbacks of the technique, ie lack of contrast, impossibility to label luminal part of organelles, iron sensitivity of some systems etc ... Nevertheless, they could have also highlighted the potential of the in cryoEM and in cryoCLEM in particular as the particles would certainly provide significant contrast.

We appreciate this comment and have added the following line to the discussion (following the list of 4 future innovations for FerriTag).

We think FerriTag may also be useful for cryoEM and cryoCLEM applications as the FerriTag particles are likely to provide significant contrast in samples prepared in this way.

Other minor points:

- In the introduction, the authors unnecessarily and wrongly criticize the invasiveness of immunolabelling. While it is clearly true for pre-embedding, it is not the case for post-embedding techniques. They should make this clearer in order to avoid defensive reaction of those readers that are still doing iEM.

Great point. We were talking about pre-embedding iEM, but we were not clear. We have changed this part to read.

Immunogold labeling has long been used for this purpose, however, pre-embedding immunogold EM is invasive and its applications are limited...

- As the authors cleverly chose clathrin as a model to prove their labelling technique, they could extend the description in giving an estimate of the labeling efficiency: can they extract this from the predicted number of site at a clathrin pit for example? I believe the stoichiometry is known from previous studies. It would be extremely interesting to estimate how many putative site (LCa) in a section and compare with the number of particles they detect.

This is a great idea but unfortunately is not feasible. The stoichiometry of clathrin cages assembled *in vitro* is known (Crowther *et al.* 1978). However, the coats found in cells are highly heterogeneous (Fig 2C in Ehrlich *et al.*, 2004). Even at high resolution the assembly of the cage is unpredictable (Fig 4 in Cheng *et al.* 2008). So, there is no way to know how many clathrins we should expect in our sections and we need to know this in order to assess labelling efficiency. On top of this, using the approach in the paper we can't be certain that every clathrin light chain has an FKBP in order to be FerriTagged.